# Serrulin: A Glycine-Rich Bioactive Peptide from the Hemolymph of the Yellow *Tityus serrulatus* Scorpion

**DOI:** 10.3390/toxins11090517

**Published:** 2019-09-06

**Authors:** Thiago de Jesus Oliveira, Ursula Castro de Oliveira, Pedro Ismael da Silva Junior

**Affiliations:** 1Special Laboratory for Applied Toxinology (LETA), Butantan Institute, São Paulo CEP 05503-900, SP, Brazil; 2Institute of Biomedical Sciences, University of São Paulo, São Paulo CEP 05508-900, SP, Brazil

**Keywords:** antimicrobial peptide, glycine-rich peptide, innate immune system, scorpions, *Tityus serrulatus*

## Abstract

Antimicrobial peptides (AMPs) are small molecules, which have a potential use as antibiotic or pharmacological tools. In chelicerate organisms, such as scorpions, these molecules constitute an alternative defense system against microorganisms. The aim of this work was to identify AMPs in the hemolymph of the *Tityus serrulatus* scorpion. Fractions of plasma and hemocytes were subjected to high-performance liquid chromatography (HPLC) and then analyzed to determine their activity in inhibiting microbial growth. One of the fractions from the hemocytes presents antimicrobial activity against microorganisms, such as Gram-negative and Gram-positive bacteria, fungi, and yeast. These fractions were analyzed by mass spectrometry, and a fragment of 3564 Da. was identified. The peptide was called serrulin, because it is derived from the species *T. serrulatus*. A comparison of the amino acid sequence of serrulin with databases shows that it has a similarity to the glycine-rich peptides described in *Cupienius salai* and *Acanthoscurria gomesiana* (spiders). Furthermore, serrulin has no hemolytic activity against human erythrocytes. While the presence of AMPs in *T. serrulatus* venom has been described in other works, this is the first work to characterize the presence of these molecules in the hemolymph (hemocytes) of this species and show its potential use as an alternative to conventional antibiotics against different species of microorganisms.

## 1. Introduction

In Brazil, there have been many studies on the species of yellow scorpion, *Tityus serrulatus*, which belongs to the Buthidae family. Scorpion stings are the most commonly reported accident of the reported accidents relating to venomous animals in Brazil, and the species *T. serrulatus* is considered one of the most dangerous species in Brazil [1]. *T. serrulatus* has a high proliferation by parthenogenesis [2] and is very common, mainly in Southeastern Brazil. *T. serrulatus* venom is composed mainly of neurotoxins that interact in molecular complexes that are essential cell membrane components (ion channels) [3]. The antimicrobial action of two *T. serrulatus* toxins (TsAP-1 and TsAP-2) and their analogs (TsAP-S1 and TsAP-S2) against microorganisms (bacteria and yeast), and their ability to inhibit the proliferation of human cancer cells [4,5] have been evaluated. While there have been studies that elucidate the composition and toxicity of the venom, there are no researches that describe the functioning of the immune system or characterize the antimicrobial peptides (AMPs) from the hemolymph (plasma and hemocytes) of *T. serrulatus*. 

The innate immune system in invertebrates is basically composed of events involving cellular responses, including phagocytosis, nodulation, and encapsulation, and humoral responses, including the production of molecules from clotting and phenoloxidase cascade and the synthesis of AMPs [6,7]. 

AMPs can be classified according to their amino acid sequence, three-dimensional (3D) structure, and action spectrum. Defensins are a group of AMPs that are amphipathic with cysteine residues, and they have patterns of conserved bonds that ensure their stability [8]. The overall similarity between the defensins of ticks and those of scorpions is up to 70%, while the similarity between defensins of scorpions and those of insects is much lower [9]. Defensins have been described in the hemolymph from three different scorpions of the species *Leiurius quinquestriatus*, which was characterized as a defensin of 4321 Da and shown to have activity against Gram-positive bacteria [10]. In the scorpion *Androctonus australis*, an androctonin molecule shows activity against Gram-negative and Gram-positive bacteria and fungi. Androctonin has two disulfide bridges and a molecular mass of 3076 Da. In the same study using *Androctonus australis*, buthinin, an antimicrobial peptide with a molecular mass of 4605 Da, and which has activity against Gram-positive and Gram-negative bacteria and similarity to a defensin described in insects (*Aeschna cyanea*), was also described [11]. The most current work describing AMPs in the hemolymph of scorpions concerns a defensin found in the *Centruroides limpidus limpidus*. This defensin is denominated Cll-dlp (*C. limpidus limpidus* defensin-like peptide), presenting activity against Gram-positive and Gram-negative bacteria, with a molecular mass of 3821 Da [12]. These studies were performed with the total hemolymph of scorpions, without separating the peptides present in the plasma and those present in the hemocytes. Therefore, it was not possible to indicate the location of these peptides.

The second AMP group contains the small open-end cyclic peptide, for example, the gomesin, isolated from the hemocytes of the spider *Acanthoscurria gomesiana*. Gomesin forms two internal disulfide bridges and adopts a β-hairpin-like fold. This peptide consists of 18 amino acids and has been tested against a variety of microorganisms [13]. The third group contains glycine-rich peptides. These AMPs are cationic and characterized by a high glycine content, for example, acanthoscurrins. These peptides are two isoforms isolated from the hemocytes of the spider *Acanthoscurria gomesiana*, and they have a positive charge and act against Gram-negative bacteria, *Escherichia coli*, and the yeast *Candida albicans* [14]. Ctenidins are glycine-rich AMPs from the hemocytes of the spider *Cupiennius salei*, and they are distinguished from acanthoscurrins by a sequence of 10 amino acids: VIDGKDDVGL. They also act against *E. coli* and *Candida albicans* [15]. 

The discovery of new AMPs from natural sources is of great importance for public health, since these molecules are pharmacological candidates due to their effective antimicrobial activity and low resistance rates [16]. AMPs have already been characterized in the hemolymph of different animals, such as acanthoscurrins and gomesin from the *Acanthoscurria gomesiana* spider [13,14], Rondonin from the *Acanthoscurria rondoniae* spider [17], Lacrain from the “centipede” *Scolopendra viridicornis* [18], and a fragment of a human fibrinogen peptide found in the hemolymph of *Triatoma infestans* [19]. All of these molecules are involved in the protection of these species and show how the discovery and characterization of bioactive peptides is very important and has a wide applicability, for example, selectivity towards cancer cells, with effects on cell proliferation [20].

In the present study, we isolated an AMP, serrulin, from the hemocytes of the scorpion *T. serrulatus*. We evaluated the antimicrobial and hemolytic activity, and mass spectrometry revealed a high percentage of glycine residues (G) and a similarity to other glycine-rich peptides from spiders.

## 2. Results and Discussion

### 2.1. Fractionation of Peptides from the Hemocytes

After extraction of the hemolymph from the *Tityus serrulatus* scorpion, hemocytes and plasma were separated by centrifugation. The acid extract from hemocytes was applied to Sep-Pak^®^ C18 cartridges for pre-fractionation. The molecules were eluted by three successive concentrations of acetonitrile (ACN) (5%, 40%, and 80%) in acidified water (0.05% TFA). All the material eluted at 40% ACN was subjected to RP-HPLC, and all fractions were tested against different microorganisms in a liquid growth inhibitory assay. The search for the hemocytes of the scorpion *T. serrulatus* showed the presence of five antimicrobial fractions (6, 2, 30, 34, and 40) (Table 1), and the numbers correspond to the fraction number from a set of fractions. 

Antimicrobial fractions from the hemocytes of the scorpion *Tityus serrulatus* present activity against microorganisms. The fractions were pre-fractionated in three concentrations of acetonitrile (ACN) (5%, 40%, and 80%) using a Sep-Pak^®^ Column, then the samples were subjected to high-performance liquid chromatography using a reverse-phase column (RP-HPLC). The fractions were analyzed in a liquid growth inhibition assay, and we found five antimicrobial fractions (6, 2, 30, 34, and 40).

Samples eluted in 5% (Figure 1) and 40% (Figure 2A) ACN, with 0.05% TFA, presented fractions with antimicrobial activity (marked with squares and/or circles). No fractions with antimicrobial activity were identified in the samples eluted in 80% ACN.

The presence of the antimicrobial peptides characterized in other arthropods and eluted at 40% ACN may be justified by the amphipathic characteristic of many antimicrobial peptides [11,12]. One of the fractions from hemocytes eluted at 40% ACN in 0.05% TFA was selected. This fraction, with antimicrobial activity and a higher absorbance, was chosen for a forward analysis. This fraction was the major peak, compared with the other fractions (Figure 2A), and it was named serrulin, referring to the species *T. serrulatus*. The homogeneity of the fraction was verified by RP-HPLC (Figure 2B). The peaks were hand-collected, and the antimicrobial activity of the major peak was reconfirmed by the inhibition of microbial growth in a liquid medium. 

The other fractions of the hemocytes that showed antimicrobial activity (fraction 6 eluted in 5% and fractions 2, 30, and 40 eluted in 40%) were submitted to the same processes but were not as homogeneous as fraction 34 (serrulin). The analyses by mass spectrometry were not as satisfactory as the results obtained for fraction 34. We believe that these fractions could be other AMPs, for example, the small open-end cyclic peptide or defensins, which have already been described in relation to other scorpions.

### 2.2. Bioassay

#### 2.2.1. Antimicrobial Activity, Minimal Inhibitory Concentration (MIC)

The antimicrobial activity was measured by liquid growth inhibition in a liquid medium [21]. The native peptide of the *T. serrulatus* scorpion hemolymph, called serrulin, was screened against a species of Gram-positive and Gram-negative bacteria, filamentous fungus, and yeast. The minimal inhibitory concentration (MIC) of serrulin was: 1.87–3.75 µg/mL (0.5–1µM) for *Micrococcus luteus* A270, 30–60 µg/mL (9–16 µM) for *Echerichia coli* SBS 363, 0.05–0.1 µg/mL (0.01–0.1 µM) for *Pseudomonas aeruginosas*, 12–24 µg/mL (3–6 µM) for *Aspergillus niger*, and 6–12 µg/mL (1.5–3 µM) for *Candida albicans* MDM8 (Table 2). Compared with the other peptides isolated in the hemolymph of scorpions, serrulin shows a higher antimicrobial activity, mainly in the low concentrations that inhibited the growth of the bacteria *Pseudomonas aeruginosas* (MIC = 0.01–0.3). The molecule acted against Gram-positive and Gram-negative bacteria, filamentous fungus, and yeast, but it was not possible to confirm if the target of the action was common for all of the microorganisms. Further experiments would be needed to elucidate the mechanisms of action of this molecule for the different tested microorganisms.

#### 2.2.2. Hemolytic Activity

No hemolytic activity of serrulin was identified at high concentrations (up to 60 µg/mL) against human erythrocytes. These results demonstrate that serrulin is not able to lyse human red blood cells (Appendix A).

### 2.3. Sodium Dodecyl Sulfate-Polyacrylamide Gel (SDS-PAGE)

Serrulin’s action against microorganisms was detected by the inhibition of microbial growth in a liquid medium, and this fraction was submitted to another step of purification. The chromatographic profile (Figure 2B) revealed a single absorbed protein peak, which was eluted at 33% ACN. The sample (20 µg) was applied to SDS-PAGE 20%, and the masses were compared to a molecular weight marker. Serrulin corresponds to a single peptide band between 3 and 6 kDa. (Figure 3), thereby confirming its homogeneity and approximate molecular mass. It is interesting to highlight that AMPs with masses of 4 kDa have been described in the hemolymph of other scorpions, such as *L. quinquestriatus* [10], *A. australis* [11], and *C. limpidus limpidus* [12].

### 2.4. Mass Spectrometry and Bioinformatics Analyses

Aliquots of samples digested (by trypsin) and not digested were analyzed separately by mass spectrometry in an LC-MS/MS, coupled with an LTQ-Orbitrap Velos. The analysis of the undigested sample revealed a single molecule of m/z 3564.0 Da (Figure 4), thus corroborating the result that the observed mass in SDS-PAGE 20% was above 3 kDa (Figure 3).

The analysis of serrulin spectra by PEAKS Studio software revealed a 3047 kDa fragment, with a 97% coverage, showing a sequence composed of 37 amino acids (GFGGGRGGFGGGRGGFGGGGIGGGGFGGGYGGGKIKG) (Figure 5), deposited in the database of transcripts of the *T. serrulatus* scorpion (midgut) [22] and its venom gland (telson) [23]. A mass near 3047 kDa was identified in this sample by the MagTran^®^ software (Figure 4), indicating that some amino acids were not identified in PEAKS Studio software. In addition, the PEAKS Studio software identified a post-translational modification (PTM) in serrulin and an amidated Lysine–Lys (K) in the C-terminal region (Figure 5). 

A BLAST (basic local alignment search tool) was used to identify similar sequences, using the UniProt as a database (Arthropod), with an E-value threshold of 10^−10^. It is relevant to highlight two groups of molecules that showed similarity to serrulin. The first result is a fragment from the secreted glycine-rich protein found in ticks of the species *Ixodes scapularis*, with an identity of 68.1% (Appendix A). Many peptides were identified, especially in the saliva of the genus *Ixodes* [24,25]. While none of these sequences corresponded to AMPs or had known activities, they were proteins obtained by a direct submission from the *Ixodes scapularis* Genome Project and did not have confirmed antimicrobial activity [26].

The second result, obtained by a database search, corresponds to two AMPs from the hemolymph (hemocytes) of the spider *Acanthoscurria gomesiana* (Theraphosidae), acanthoscurrin 1 (Acantho 1) and acanthoscurrin 2 (Acantho 2). Both are molecules classified as glycine-rich antimicrobial peptides. Serrulin has a similarity of 63% to Acantho 2. The alignment of serrulin with the last 37 amino acids of Acantho 1 and Acantho 2 shows a conserved sequence of amino acids in the C-terminal region (Figure 6). It is very interesting to note that, in both, serrulin and acanthoscurins have an amidation of the lysine residue in the C-terminal. Acanthoscurrins has a marked activity against *C. albicans* (MIC = 1.15–2.3 µM) and *E. coli* (MIC = 2.3–5.6 µM) [14]. Despite the similarity in the glycine content and activity of serrulin and acanthoscurrins, serrulin presents remarkable differences from acanthoscurrins, for example, the mass of serrulin is 3564.0 Da, while that of Acantho 1 and Acantho 2 is 10169 Da and 10225 Da, respectively. In addition, serrulin’s activity against *M. luteus* and *A. niger* could be detected. It is interesting to note that there is an amidation of the lysine residue in the C-terminal in both serrulin and acanthoscurins (Figure 5), suggesting that this modification may be important for the activity of these molecules. 

Ctenidins are antimicrobial glycine-rich peptides from the hemocytes of the mature female spider *Cupiennius salei*. Three molecules were isolated and denominated ctenidins 1–3. The monoisotopic molecular masses were 8810, 9507, and 9564 Da. Ctenidins presented antimicrobial activity against *E. coli* (MIC = 2.5–5 µM), and, with ctenidins, the growth of *Staphylococcus aureus* was reduced but not inhibited at concentrations of up to 10 µM. The amidation of the C-terminal of the mature peptide ctenidin is an ordinary modification of some AMPs from other organisms, such as chelicerates and crustaceans [27,28,29], making these peptides more stable. In addition, the amidation is present in both acanthoscurrins and serrulin. 

In *T. serrulatus* transcripts databases, the fragment corresponding to serrulin was found to be a “protein not characterized”. This is because the identification of the molecules is directly related to a structure research (midgut and venom gland), and serrulin is not a toxin or a protein involved in digestion. However, it may be present in these structures, because it is present in the hemolymph (whole body) of this species.

Analyses using the ExPASy ProtParam [30] tool indicate that serrulin is, like other AMPs, a cationic molecule, containing a positive charge (+4) (Appendix A). Generally, cationic antimicrobial peptides act on the membrane of microorganisms by an electrostatic difference [31], but there are AMPs that act internally and can interact on ribosome, internal proteins, or nucleic acids (DNA or RNA) [32,33,34]. Like ctenidins and acanthoscurrins, the mechanism of action of serrulin was not identified in this work, so it is not possible to compare its characteristics with those of similar molecules.

The purification processes (Sep-Pak and HPLC), biological activities (antimicrobial activity, MIC, and hemolytic assay), characterization (SDS-Page), and computational data acquisition (mass analysis) yielded quantities of the native molecule that allowed us to initially identify and characterize serrulin. To elucidate its mechanisms of action, it would be necessary to extract more hemolymph and perform the purification processes again.

It would be interesting to identify the site and mode of action of serrulin in the different microorganisms, but due to the difficult process of obtaining this molecule, we intend to carry out such studies with synthetic sequences of its conserved regions in the future.

## 3. Conclusions

We isolated and characterized an antimicrobial peptide from the hemolymph of the *Tityus serrulayus* scorpion, and we named this peptide serrulin, referring to the name of the species. Serrulin exhibits a similar primary structure to glycine-rich antimicrobial peptides, which have already been described in relation to other arachnids, such as ticks and spiders, and it has a molecular mass of 3564 Da. Serrulin shows antimicrobial activity against Gram-positive and Gram-negative bacteria, filamentous fungi, and yeast. However, serrulin does not present cytotoxicity against human erythrocytes at the tested concentrations. In this work, the first antimicrobial peptide from the hemolymph of the yellow scorpion, *Tityus serrulatus*, was described. This acts on the innate immune system of the scorpion to aid in its self-protection. Moreover, the study of this molecule can be of great interest for the discovery of a new antibiotic molecule that acts against resistant bacteria or a pharmacological tool.

## 4. Material and Methods

### 4.1. Animals and Hemolymph Extraction

Scorpions from the species *Tityus serrulatus* (Chelicerata, Buthidae) were collected in São Paulo and the Minas Gerais states and kept alive in a vivarium in the Special Laboratory for Applied Toxinology, under the licenses: Permanente Zoological Material Licenses 11024-3 IBAMA and Special Authorization for Access to Genetic Patrimony: 010345/2014-0, Instituto Butantan, São Paulo, Brazil. 

This project was accepted by the Commission of Ethics regarding the use of Animals of the Butantan Institute (Comissão de Ética no Uso de Animais do Instituto Butantan), number: 1839260716, on 5 November 2018. The total hemolymph (about 1 mL) was extracted from 50 individuals by cardiac puncture using a hypodermic needle. The hemolymph was extracted with a sodium citrate buffer (0.45 M NaCI; 0.1 M glucose; 30 mM trisodium citrate; 26 mM citric acid; and 10 mM EDTA), pH 4.6, to prevent hemocyte degranulation and clotting. The separation of the plasma and hemocytes was achieved through centrifugation in 800× *g* for 15 min at 4 °C. The hemocytes were acidified in 5 mL of acetic acid 2 M (Synth, Diadema, Brazil), homogenized using a Dounce^®^ homogenizer, and maintained under agitation on ice for 30 min. Then, the sample was centrifuged at 16,000× *g* for 10 min at 4 °C. The plasma was acidified in 20 mL of acidified ultrapure water [trifluoroacetic acid (TFA) 0.05%]. The solution was maintained under agitation on ice for 30 min, and the sample was centrifuged at 16,000× *g* for 10 min at 4 °C [35].

### 4.2. Fractionation of Antimicrobial Peptides

The supernatant of the plasma and hemocytes (separated) was loaded in Sep-Pak^®^ C18 cartridges (Waters Corp., Milford, MA, USA), and the elution was performed with acetonitrile (ACN) (J.T.Baker^®^; Avantor Performance Materials, Center Valley, PA, USA) at different concentrations (5%, 40%, and 80%) and equilibrated in 0.05% TFA. The fractions were concentrated in a vacuum centrifuge (SpeedVac^TM^ Savant^TM^; Thermo Fisher Scientific, Waltham, MA, USA) and reconstituted with 1 mL of ultrapure water. Reverse-phase high-performance liquid chromatography (RP-HPLC) was carried out at room temperature on a Shimadzu LC-10 HPLC system. The column was a semi-preparative Jupiter^®^ RP-C18 LC-Column (10 µm, 300 Å, 250 × 10 mm) (Phenomenex International, Torrance, CA, USA). The elution was performed with a linear ACN gradient, equilibrated in 0.05% TFA for 60 min at a flow rate of 1.5 mL/min (0–20% for the fraction eluted in 5%, 2–60% for the fraction eluted in 40%, and 20–80% for the fraction eluted in 80%). The absorbance was monitored at 225 nm and 280 nm, and the fractions were hand-collected. Then, lyophilized fractions were suspended in 300 µL ultrapure water for antimicrobial assays. 

The molecule described in this research, faction 34, was subjected to a second step of fractionation using a linear gradient of ACN (13–43%) with 0.05% TFA for 60 min at a flow rate of 1.0 mL/min, and an analytic column Jupiter^®^ C18 LC-Column (10 µm 300 Å, 250 × 4.60 mm) (Phenomenex International, Torrance, CA, USA), with ACN equilibrated in 0.05% TFA. The fractions were hand-collected.

### 4.3. Bioassays

The antimicrobial activity was monitored by a liquid growth inhibition assay against Gram-negative and Gram-positive bacteria, fungi, and yeast, provided by Dr. Pedro Ismael da Silva Junior at the Instituto Butantan, Brazil. Briefly, a suspension of the microorganisms collected in the mid-logarithmic growth phase was used. Bacteria were cultured in a medium containing a broth with a poor nutrient content (PB) (1.0 g peptone in 100 mL of water containing 86 mM NaCl at pH 7.4; 217 mOsm), and the fungi and yeast were cultured in a poor potato dextrose broth (1/2-strength PDB) (1.2 g potato dextrose in 100 mL of water at pH 5.0; 79 mOsm). The assay was carried out using 80 µL of the microorganism suspension in 96-well sterile microtiter plates, and 20 µL of the fraction was collected by HPLC, until the final volume of 100 µL was obtained (5 × 10^4^ microorganisms). The plates were incubated for 18 h and 24 h for bacteria and fungi/yeast, respectively, at 30 °C. Sterile water was used as a microbial growth control, and streptomycin or tetracycline was used as a control of the growth inhibition. Antimicrobial activity was determined by measuring the absorbance at 595 nm. 

The minimum inhibitory concentration (MIC) was determined against microorganisms, as described above, using a serial dilution of serrulin. MICs are expressed as the interval of the concentration [*a*]–[*b*], where [*a*] is the highest tested concentration at which the bacteria grow, and [*b*] is the lowest concentration which causes a growth inhibition of 100% [21].

The hemolytic activity was tested using human erythrocytes from a healthy adult donor. The Ethics Committee was the University of Sao Paulo School of Medicine (USP) FMUSP, n° 17526619.3.0000.0065. Cells were first washed three times in a phosphate-buffered saline (PBS) (35 mM phosphate buffer, 0.15 M NaCl, pH 7.4) by centrifugations at 700× *g* for five minutes. Then, a suspension of 3% (*v*/*v*) of washed erythrocytes was incubated with 50 µL of serrulin and PBS in a 96-well plate (assays were conducted in triplicate) for 3 h at 37 °C (with a final volume of 100 µL). The supernatant absorbance was measured at 405 nm. The hemolysis percentage was expressed for a 100% lysis control (erythrocytes incubated with 0.1% Triton X-100, (Sigma-aldrich, St. Louis, MO, USA), and PBS was used as a negative control [36].

### 4.4. Sodium Dodecyl Sulfate-Polyacrylamide Gel (SDS-PAGE)

Sodium dodecyl sulfate-polyacrylamide gel electrophoresis (SDS-PAGE) was carried out, and proteins were stained with Coomassie blue R [37]. Thirty micrograms of serrulin, solubilized in 20 µL of ultrapure water, and a sample buffer were submitted to electrophoresis under reducing (data not shown) and non-reducing conditions on 20% SDS-polyacrylamide gels, using invitrogen SeeBlue^®^ (Life Technologies do Brasil, Itapevi, SP, Brazil ) as a molecular weight marker.

### 4.5. Enzymatic “in Gel” Digestion 

The selected protein (serrulin) was cored from the gels and placed in a microcentrifuge tube. The band was de-stained in 0.5 mL 50% methanol/5% acetic acid overnight at room temperature, before dehydration in 200 µL acetonitrile, followed by complete drying in a vacuum centrifuge. The proteins were reduced by the addition of 50 µL Dithiothreitol-DTT (10 mM) and alkylated by the addition of 50 µL iodoacetamide-IAA (100 mM) (both for 30 min at room temperature). The band was dehydrated in 200 µL acetonitrile and hydrated in 200 µL ammonium bicarbonate (100 mM). The dehydrated band was dried completely in a vacuum centrifuge and rehydrated in 50 µL of 20 ng µL^−1^ ice-cold porcine trypsin (Sigma-aldrich^®^). The digestion was carried out overnight at 37 °C. The tryptic peptides produced in the digestion were collected by successive extractions with 50 µL (50 mM ammonium bicarbonate) and 50 µL (50% acetonitrile/5% formic acid (twice)). The total extract was concentrated in a vacuum centrifuge [38].

### 4.6. Mass Spectrometry and Bioinformatics Analyses

The naive and tryptic fragments of serrulin were resuspended in formic acid (0.1%) and injected into an EASY-nL II nanoflow liquid chromatography system (Thermo Fisher Scientific), in tandem with an LTQ-Orbitrap Velos mass spectrometer (Thermo Fisher Scientific), at a flow rate of 200 nL/min and with a linear gradient (from 0% to 80% mobile phase B) (0.1% formic acid in 100% acetonitrile) for 40 min. Tryptic peptides were separated by a capillary reverse-phase C18 fritted-tip analytical column (ID 75 μm × OD 360 μm, 10 cm length) and in-house packed with 5 μm Aqua C18 (125 Å, Phenomenex, Torrance, CA, USA). The spectrometer was operated in a positive mode, and the 10 most intense peaks were selected for collision-induced dissociation (CID) fragmentation, after acquiring each full scan. For the fragment scans, the settings were: An isolation window of 2 Da; a max list size of 500; a time window of 30 s; a minimum signal of 5000; an activation time of 10 ms; and a normalized collision energy of 35% [39].

The raw data files (Xcalibur Raw File) were used in a PEAKS Studio software (version 8.5, Bioinformatics Solution, Waterloo, ON, Canada) search. The analysis involved a 10-ppm error tolerance for the precursor ions, and 0.6 Da for the fragment ions. Oxidation was considered a variable modification, and trypsin was considered enzyme-specific. The MS/MS spectra were searched against a transcriptomic midgut database of *T. serrulatus* (downloaded in FASTA format on 20 August 2018, from NCBI UniProtKB; http://www.uniprot.org/; 44,111 entries). The obtained amino acid sequence was subjected to a BLAST search against the GenBank (NCBI) and UNIPROT databases using the alignment tool Protein BLAST P (Target database = Arthropoda, E-Threshold = 10, Matrix = Auto, Gapped = yes, Hits = 250). For mass analysis by the deconvolution of the ions, the MagTran^®^ Program, version 1.02, was used [40].

The physico-chemical parameters of serrulin (such as the total number of positively and negatively charged residues, molecular weight, and theoretical pI) were calculated using the ProtParam tool, available through the portal ExPASy of the Swiss Institute of Bioinformatics website (https://web.expasy.org/protparam/) [30].

## Figures and Tables

**Figure 1 toxins-11-00517-f001:**
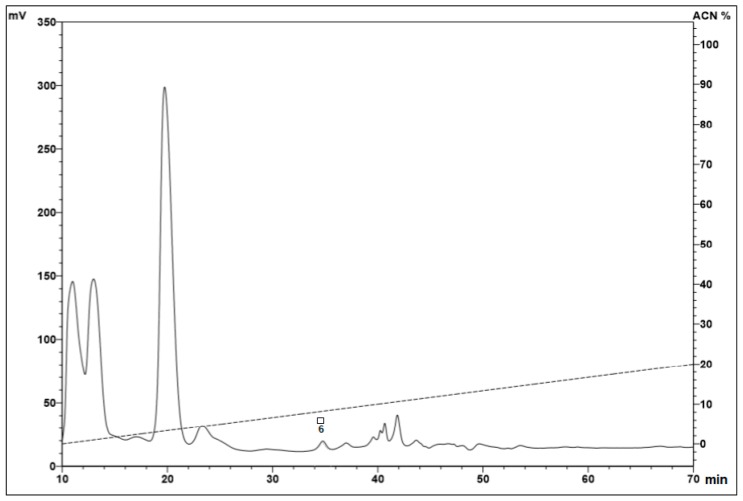
Reverse-phase high-performance liquid chromatography (RP-HPLC) of the hemocyte samples eluted at 5% acetonitrile (ACN). After the fractionation step using a Sep-Pak^®^ C18 column, the samples eluted in 5% ACN were subjected to a Jupiter^®^ C18 semi-preparative column, with a linear gradient from 0% to 20% ACN (dotted line) in acidified water for 60 min (1.5 mL/min). The fractions were collected manually and submitted to a test of the inhibition of microbial growth in a liquid medium. Peaks with squares indicate the antimicrobial activities against *Aspergillus niger* (□).

**Figure 2 toxins-11-00517-f002:**
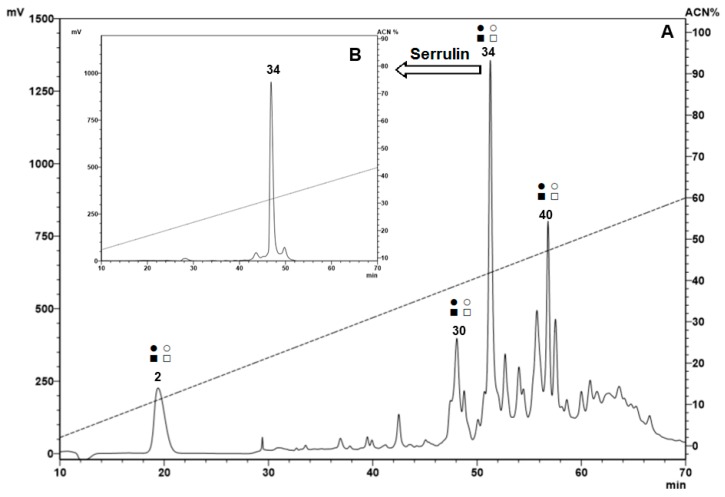
Reverse-phase high-performance liquid chromatography (RP-HPLC) of the hemocyte samples eluted at 40% acetonitrile (ACN). After the fractionation step using a Sep-Pak^®^ C18 column, the samples eluted in 40% ACN were subjected to a Jupiter^®^ C18 semi-preparative column, with a linear gradient from 2% to 60% ACN (dotted line) in acidified water for 60 min. The fractions were collected manually and submitted to a test of the inhibition of microbial growth in a liquid medium. Peaks with squares and circles indicate antimicrobial activities against the tested microorganisms: *Escherichia coli* SBS 363 (●), *Microccocus luteus* A270 (○), *Candida albicans* MDM 8 (■), and *Aspergillus niger* (□) (**A**). Serrulin, indicated with an arrow, was submitted to RP-HPLC using a Jupiter^®^ C18 analytic-column, with a linear gradient from 13% to 43% ACN (dotted line) in 0.05% TFA, for 60 min. The fractions were collected manually and submitted to a test of the inhibition of microbial growth in a liquid medium and mass analyses (**B**).

**Figure 3 toxins-11-00517-f003:**
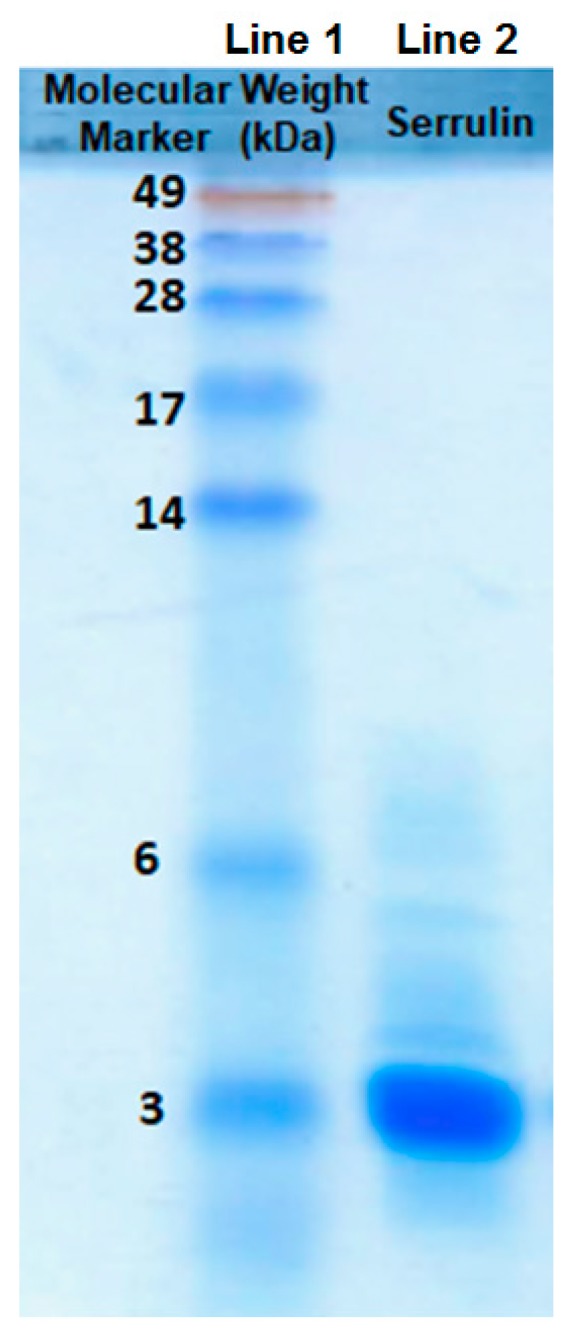
Sodium Dodecyl Sulfate Polyacrylamide Gel Electrophoresis 20% (SDS-PAGE-20%), stained with Coomasie-R Blue. Line 1, molecular weight marker (invitrogen SeeBlue^®^), expressed in kDa; line 2, serrulin in a non-reducing condition (20 µg).

**Figure 4 toxins-11-00517-f004:**
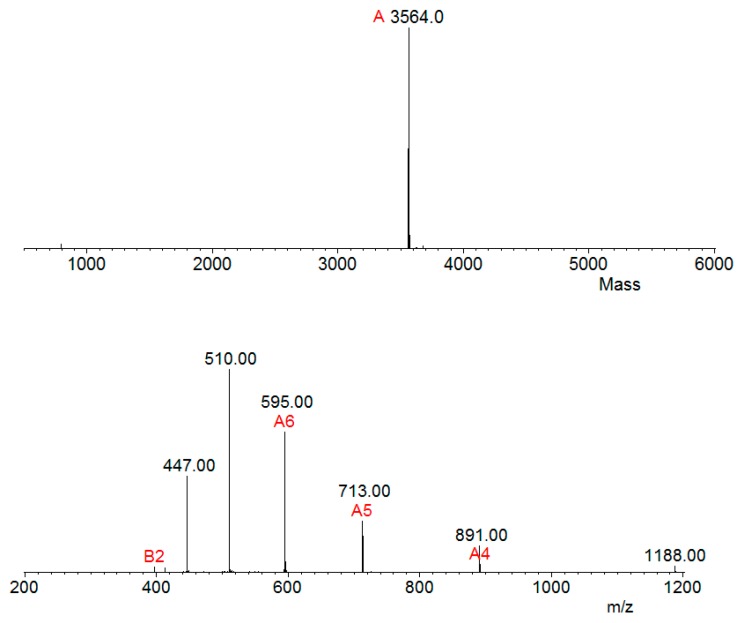
Serrulin spectrum. Mass spectrometry analyses (LC-MS/MS, coupled with a LTQ-Orbitrap Velos) of the peptide serrulin revealed an m/z of 3564.0 Da. Ions were submitted to MagTran^®^ software.

**Figure 5 toxins-11-00517-f005:**
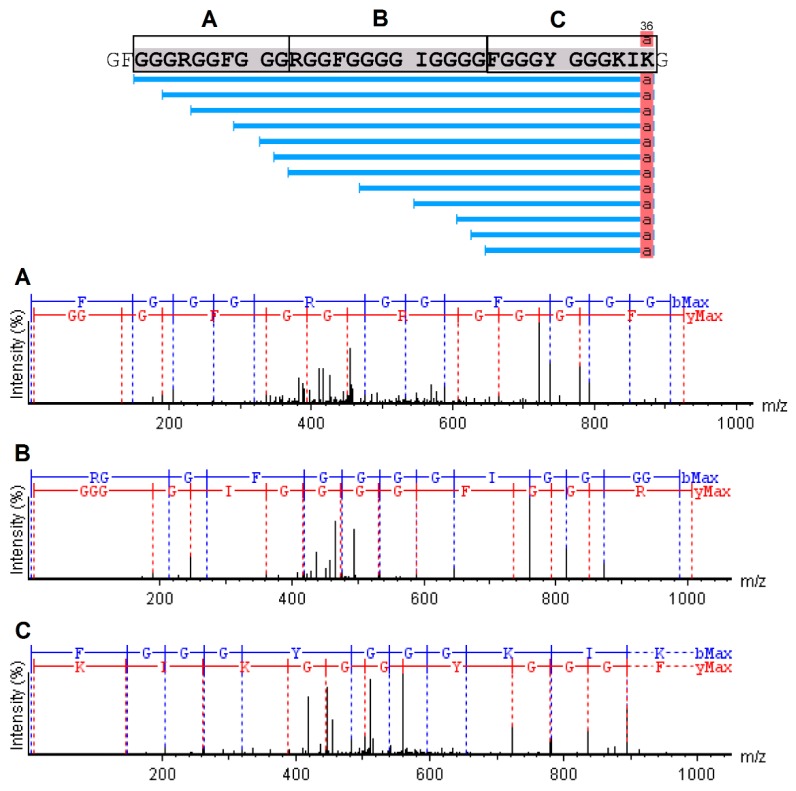
Collision-induced dissociation (CID) spectrum of the novo sequence from serrulin. The ions belonging to the -y (red) and -b (blue) series, indicated in the spectrum, correspond to the amino acid sequence of the peptide: GFGGGRGGFGGGRGGFGGGGIGGGGFGGGYGGGKIKG. The fragments of the sequenced peptide are represented by standard amino acid code letters.

**Figure 6 toxins-11-00517-f006:**
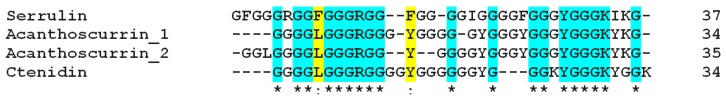
Alignment of the amino acid sequence of serrulin with other antimicrobial glycine-rich peptides. Using the UNIProt alignment, the serrulin sequence was aligned with AMPs: Acanthoscurrin 1, Acanthoscurrin 2, and Ctenidin. Amino acids that were identical in all sequences are shown in blue (or *), and minor modifications are shown in yellow or (:).

**Table 1 toxins-11-00517-t001:** Antimicrobial activity fractions from hemocytes.

Microorganism					
Hemocytes	Fractions	*Escherichia coli* SBS363	*Micrococcus luteus* A270	*Candida albicans* MDM8	*Aspergillus niger*
5%	6	–	–	–	+
	2	+	+	+	+
	30	+	+	+	+
40%	34	+	+	+	+
	40	+	+	+	+

**Table 2 toxins-11-00517-t002:** Antimicrobial activity spectrum of serrulin.

Strain	MIC (µg/mL)	µM
**Gram-positive bacteria**		
*Micrococcus luteus* A270	1.87–3.75	(0.5–1)
**Gram-negative bacteria**		
*Escherichia coli* SBS 363	30–60	(9–16)
*Pseudomonas aeruginosa* ATCC 27853	0.05–0.1	(0.01–0.3)
**Fungi**		
*Aspergillus niger*	12–24	(3–6)
**Yeast**		
*Candida albicans* MDM8	6–12	(1.5–3)

MIC, minimum inhibitory concentration (µg/mL): µM. The MIC refers to the concentration necessary to achieve a growth inhibition of 100%.

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
