# Peer review of "Serrulin: A Glycine-Rich Bioactive Peptide from the Hemolymph of the Yellow Tityus serrulatus Scorpion"

_toxins, 2019, doi:10.3390/toxins11090517_

Round 1

Reviewer 1 Report

After reading the submitted manuscript entitled: “Serrulin a glycine-rich bioactive peptide from the hemolymph of the yellow scorpion Tityus serrulatus” I have the following comments to make. They will also be sent to the authors.

 It is always welcome the discovery of a new natural molecule with biological activities, especially in the world of antimicrobial peptides since they have a mode of action different to traditional and current antimicrobial drugs. In this work, the authors show a wide variety of experiments to extract, purify, analyze and test for activity a new antimicrobial peptide (serrulin) with a glycine-rich sequence. These experiments look appropriate for the characterization of such peptide but the way they are described and presented need some review. Here are my notes and suggestions:

Sometimes the text reads like translated by an automatic computer translator. The experiments are fine but I have noticed I was paying more attention to the writing than to the results and I had to read the manuscript twice to get the introduction, results and a discussion, yet, I still got a bit confused.

 I would suggest having a native English speaker to read the whole text and make corrections where necessary. Also, to put some order on the information mentioned in the introduction especially on the last paragraph (lines 61 to 71). The authors mention a lot of “important molecules” without any particular reason except that it shows what is done, which is fine, but the way it is narrated is confusing. I would suggest to review the whole text and, as mentioned, put some order in the information presented.

Just as a few examples of grammatical issues:

There are some text/language issues starting as early as line 8:

- Line 8: “AMPs from the hemolymph oh the scorpions”. I think it should say: “AMPs from the hemolymph of the scorpions”

- Lines 12-13:” This fraction was analyzed by mass spectrometry and a fragment of 3564 was identified.1 Da.” Is the fragment of 3564 1 Da or it was fragment 3564 one of 1 Da? Suggested explanation.

- Line 29: “The AMPs can be classified…” suggest to say: “AMPs can be classified…”

- Line 35: “  …which has an action against 35 bacteria Gram-negative, Gram-positive and against fungi.” Suggest to say: “… which has (or shows) activity against 35 bacteria Gram-negative, Gram-positive and against fungi.”

- Line 36: “The androctonin has two bridges sulfide and molecular mass of 3076 Da” I suggest to say: “Androctonin has two sulfide bridges and a molecular mass of 3076 Da”

- All over the text: It’s not common in English to show the article THE before personal names like The AMPs, The defensins (line 30), The buthinin  (line 37), The Gomesin (line 46), The serrulin (line 141), etc.  I would suggest to review this.

I can go on with all these (and others) grammatical issues but I think I will stop here because they are shown all over the text.

- On line 48 the text mentions “The third group is the Glycine-Rich Peptides…” but I try to see where the first and second groups are. I think the first group is defensins and the second is “the small Open-End Cycli peptide…” (line 45) but I think it needs to be clarified.

- Lines 74-75: “hemocytes and plasma were separated by centrifugation as previously described.” It needs a reference.

- Line 76-77: “The molecules were eluted by three successive concentrations of acetonitrile-ACN (5%, 40% e (and?) 80%)”. Is it acetonitrile different to ACN? Better shown in line 84.

- Table 1 (lines 81 to 82): I suggest to clarify what the numbers 6, 2, 30, 34 and 40 are. I guess they are retention times form HPLC? Are they the fraction number from a set of fractions?

- Lines 91 and 92: 05% acetonitrile? Is it 0.5 % instead? Or just 5%?

- Line 136: “Serrulin is a molecule isolated from the hemocytes of the scorpion T. serrulatus by RP-HPLC” Probably it is no necessary to repeat what we already know in detail just two sections above.

Author Response

To Reviewer,

 First of all, I would like to thank you, all the suggestions made to improve the work. Really, writing in English is a challenge for me. All your comments and suggestions were extremely clear and timely. I agreed and made all the changes suggested in the grammar examples.

I would suggest having a native English speaker to read the whole text and make corrections where necessary. Also, to put some order on the information mentioned in the introduction especially on the last paragraph (lines 61 to 71). The authors mention a lot of “important molecules” without any particular reason except that it shows what is done, which is fine, but the way it is narrated is confusing. I would suggest to review the whole text and, as mentioned, put some order in the information presented.

First point: "put some order on the information mentioned".

I made some changes in the introduction.

Before: I started the text, talking about the immune system, AMPs, Tityus serrulatus, all the work in our laboratory and ended with the purpose of the research.

I modified the order for: Tityus serrulatus, immune system, AMPs, works carried out with the characterization of only AMPs in hemolymph of some arthropods and lastly I justify the research.

I believe that this order has clarified the text.

Second point: Before submitting the text we had sent the article to a company that corrects English and we paid for it. After receiving the comments from the reviewers we contacted the company again, who agreed to do the review at no additional cost. Currently the article is being analyzed by the translation and correction company.

Best Regards,

Reviewer 2 Report

This manuscript reports the first characterisation of an antimicrobial peptide from the hemolymph of the scorpion Tityus serrulatus. The present work provides further insight into scorpion hemolymph and its components. The following concerns needs to be addresses:

Major

- Figure 2: It can be regretted that the other peak which also showed inhibition of microbial growth are not further characterised and the active AMP identified. Do the authors expect to find in these fractions glycine rich peptides as well? The manuscript would benefit greatly if the other fraction were to be explored further.

-Extensive English editing is required for this manuscript. Although the results are scientific sound, it is often difficult to read and understand the text due to poor English. Some example:

Line 36: Gram-negative and Gram-positive bacteria

Line 36: two disulfide bridges

Line 37: and a molecular mass

Line 40 …is about

Line 96: … and eluting..

Line 103: … the activity was reconfirmed (instead of the word again)

Line 177: … are proteins

Line 181: rewrite sentence

Minor

Figure 5: there is no need to make this a full page figure. The size should be reduced and the resolution enhanced.

Author Response

To Reviewer,

Firstly, I would like to thank all the comments and suggestions, they were really very welcome.

All specific suggestions were accepted and changed in the text.

- Figure 2: It can be regretted that the other peak which also showed inhibition of microbial growth are not further characterized and the active AMP identified. Do the authors expect to find in these fractions glycine rich peptides as well? The manuscript would benefit greatly if the other fraction were to be explored further.

We believe that these other fractions may be other AMPs, such as defensins found in hemolymph of other scorpions (this information I included in the text after your comment). But we haven't stopped the analysis of the other fractions with antimicrobial activity.

-Extensive English editing is required for this manuscript. Although the results are scientific sound, it is often difficult to read and understand the text due to poor English.

Before submitting the article we had sent the text to a company that corrects English, we paid for it. After the suggestions submitted by the 2 Reviewers, we contacted the company for a complaint, but they will do the review again (no additional costs), we are waiting for the return.

Figure 5: there is no need to make this a full page figure. The size should be reduced and the resolution enhanced.

Perfect, I did not know this function in the MagTran software, but there was the option to increase the font so I did, once again thank you very much.

Best Regards,